First use of molecular evidence to match sexes in the Monstrilloida (Crustacea: Copepoda), and taxonomic implications of the newly recognized and described, partly Maemonstrilla-like females of Monstrillopsis longilobata Lee, Kim & Chang, 2016

Jeon Donggu 1
Lim Donghyun 2
http://orcid.org/0000-0002-9873-1033 Lee Wonchoel 1 wlee@hanyang.ac.kr
Soh Ho Young 3 hysoh@chonnam.ac.kr
1 Department of Life Science, College of Natural Sciences, Hanyang University , Seoul , South Korea
2 Jeju Branch Office, Korea Fisheries Resources Agency (FIRA) , Jeju, Jeju-do , South Korea
3 Faculty of Marine Technology, College of Fisheries and Ocean Science, Chonnam National University , Yeosu, Jeollanam-do , South Korea
Costello Mark
Electronic publication date: 2018 Jun 13
Publication date: 2018
Volume: 6
Electronic Location ID: e4938
Received 2018 Apr 6; Accepted 2018 May 21
Copyright: © 2018 Jeon et al.
Copyright year: 2018
Copyright holder: Jeon et al.
License: This is an open access article distributed under the terms of the Creative Commons Attribution License, which permits unrestricted use, distribution, reproduction and adaptation in any medium and for any purpose provided that it is properly attributed. For attribution, the original author(s), title, publication source (PeerJ) and either DOI or URL of the article must be cited.
License URL: https://creativecommons.org/licenses/by/4.0/

Keywords: COI primer, Korea, Mitochondrial and nuclear genes, Sexual dimorphism, Taxonomy, Planktonic copepods, Antennular nomenclature

Funding: National Institute of Biological Resources Ministry of Environment of the Republic of Korea NIBR201501201 BK21 plus program (Eco-Bio Fusion Research Team) 22A20130012352 National Research Foundation This work was supported by a grant from the National Institute of Biological Resources (NIBR), funded by the Ministry of Environment of the Republic of Korea (NIBR201501201) and by the BK21 plus program (Eco-Bio Fusion Research Team, 22A20130012352) through the National Research Foundation (NRF) funded by the Ministry of Education of Korea. The funders had no role in study design, data collection and analysis, decision to publish, or preparation of the manuscript.

==============================
Two forms of the monstrilloid copepod, one represented by males, the other by females, were dominant among the monstrilloids collected while sampling zooplankton with a light trap in southern coastal waters of Korea. Morphological examination revealed that the males are conspecific with the previously reported and rather specialized Korean species Monstrillopsis longilobata Lee, Kim & Chang, 2016, hitherto known only from males that have extremely long genital lappets. The females also show several diagnostic features of Monstrillopsis, such as prominent eyes, bilobed fifth legs with the inner lobe unarmed and reduced, and four urosomal somites. In addition, though, these females are extraordinary among all Monstrilloida in that their ovigerous spines are directed ventrally, not posteriorly as in most species or anteriorly as in those assigned to Maemonstrilla. Genetic divergence analyses based on partial mitochondrial cytochrome c oxidase subunit I, complete Internal Transcribed Spacer 1 (ITS1)–5.8S–ITS2, and partial 28S rRNA revealed little or no genetic divergence between the present males and females, thus demonstrating that they are mutually conspecific. The present report (1) provides the first morphological description of female M. longilobata with the proposal of a revised nomenclature for antennular setal armament; (2) presents the molecular evidence for conspecificity of the males and females; and (3) lists several morphological characteristics that are sexually dimorphic in this species, and thus likely also in other monstrilloids. Matters bearing on the validity of the genera Haemocera, Monstrillopsis, and Maemonstrilla are discussed.

Introduction

The order Monstrilloida Sars, 1901 is distinctive among copepods both morphologically and ecologically. Monstrilloids have a protelean life history that includes an endoparasitic juvenile phase and a planktonic adult phase. The early infectious nauplii are free-living but soon infect several kinds of marine invertebrates such as polychaete worms, gastropod and bivalve molluscs, and sponges (Boxshall & Halsey, 2004; Huys et al., 2007; Suárez-Morales et al., 2014). Details of the endoparasitic stages are still unclear, but they probably undergo several copepodite instar molts during the process of development (Malaquin, 1901; Raibaut, 1985; Suárez-Morales et al., 2014). The free-swimming adults are non-feeding, with no oral appendages or second antennae, and seem to be only reproductive forms. These reductions of mouth appendages including the antennae and almost similar patterns of swimming legs of the species of the Monstrilloida have the effect of making most monstrilloid species morphologically similar, and together with the paucity of knowledge about their biology, frequently cause problems in species differentiation and identification.

According to Suárez-Morales (2011), about 120 species were then known in the family Monstrillidae Dana, 1849, and only 21 of them (18% of the total) had been reported from both sexes whereas 95 species (82%) were known only from a single sex. Currently about 160 species have been recognized on the strength of a large body of taxonomic works, especially studies like those of Suárez-Morales, Bello-Smith & Palma (2006), Grygier & Ohtsuka (2008), and Suárez-Morales & McKinnon (2014, 2016). These copepods are currently classified into six supposedly valid genera: Monstrilla Dana, 1849, Cymbasoma Thompson, 1888, Monstrillopsis Sars, 1921, Maemonstrilla Grygier & Ohtsuka, 2008, Australomonstrillopsis Suárez-Morales & McKinnon, 2014, and Caromiobenella Jeon, Lee & Soh, 2018.

Despite recent taxonomic advancements, the species newly reported during the years 2011–2018 were still mostly each based on a single sex. Of the 48 newly described species in six genera reported during that period, only four are known from both sexes (Razouls et al., 2005–2018; Walter & Boxshall, 2018). In the extreme case of Maemonstrilla, all 11 species are known only from females. To shed more light on monstrilloid taxonomy, the confirmation of sexual counterparts and the provision of new morphological criteria for genus and species recognition are still needed. Matching the two sexes seems not to be an easy task. One main obstacle is the high degree of sexual dimorphism (Suárez-Morales, 2007; Suárez-Morales, Ramírez & Derisio, 2008), which makes some of the traditional and conventional methods, especially those dependent on morphological characters, unreliable. Methods based on co-occurrence of both sexes and their utilization of a common host may also fail to guarantee perfect pairing (Grygier & Ohtsuka, 2008; Suárez-Morales, 2011; also see Discussion).

The use of molecular markers is likely to be far the most effective and definitive method for matching male monstrilloids to their corresponding females (Suárez-Morales, 2011; Suárez-Morales et al., 2017). With the great advances in molecular technology over the past several decades, much nucleotide sequence data have become available and also been practically applied to various purposes such as DNA-based taxonomy and DNA barcoding (Vogler & Monaghan, 2007; Blanco-Bercial et al., 2014). DNA-based taxonomy, which typically involves species circumscription and delineation, has been widely used in various taxa, and many gene markers have been developed and applied accordingly (Vogler & Monaghan, 2007; Bucklin, Steinke & Blanco-Bercial, 2011). The application of molecular tools to monstrilloid research has been rare, and only 129 sequence search results based on the keyword “Monstrilloida” could be obtained from GenBank (accessed on May 2, 2018). A few previous studies such as Huys et al. (2007) and Baek et al. (2016) have utilized genetic information obtained from monstrilloids, but only at the level of the order; genus and species relations remain uncertain.

In this new study, we demonstrate the conspecificity of individuals by using both morphological and molecular evidence. The two sexes of Monstrillopsis longilobata Lee, Kim & Chang, 2016 show extreme sexual dimorphism in their main features, but microcharacters such as pore patterns display potential homologies. To confirm the validity of such morphological matching of the sexes, sequences of three gene markers, viz., the region coding for partial mitochondrial cytochrome c oxidase subunit I (mtCOI), the nuclear multigene region consisting of partial 18S ribosomal RNA (rRNA)–Internal Transcribed Spacer 1 (ITS1)–5.8S rRNA–ITS2–partial 28S rRNA (called ITS1–5.8S–ITS2 hereinafter), and an additional region of partial 28S rRNA were used for analyses of genetic divergence. Mitochondrial DNA sequences, known to characterize a fast-evolving gene (Blanco-Bercial, Bradford-Grieve & Bucklin, 2011; Willett, 2012), are the most frequently used sequences for genetic and phylogenetic investigations. In contrast, nuclear genes with a relatively small number of genetic mutations have generally been judged inadequate for species delimitation. Machida & Tsuda (2010), however, noted various errors that might arise from analyses based only on mitochondrial genes; problems that could be caused by the existence of nuclear mitochondrial pseudogenes, the occurrence of mitochondrial introgression, and the pattern of decent via maternal inheritance (also see Blanco-Bercial et al., 2014). We therefore consider it worthwhile to include the results from nuclear gene sequences, for the sake of any support they may provide for the current results based on mitochondrial genes.

Materials and Methods

Sample collection and treatment for morphological analyses

A hand-made light trap was used, consisting of a 400 mm long PVC pipe with a mouth diameter of 100 mm. A conical funnel was attached within one end, and the other end was completely closed with a cap equipped a light-emitting diode (LED) flashlight of 110 lumens (Kovea, Incheon, Korea) as a light source. After use, the trap’s contents were filtered through a sieve of 63 μm mesh and the retained material, including copepods, was immediately washed several times with 99.5% ethanol, which was replaced with newly prepared 99.5% ethanol upon arrival at the laboratory. All samples were stored in a refrigerator at 4 °C and monstrilloids were sorted out later under a SMZ645 stereomicroscope (Nikon, Tokyo, Japan).

Specimens of M. longilobata were first examined as whole mounts on depression slides using 0.25–0.50% sodium phosphate tribasic dodecahydrate (Na3PO4·12H2O) (Daejung, Siheung, Korea) solution as the slide mountant (Van Cleave & Ross, 1947; Huys & Boxshall, 1991) in order to restore their original shape. Drawings were made using an Eclipse 80i compound microscope (Nikon, Tokyo, Japan) with differential interference contrast optics and a drawing tube. After the observation of habitus, a specimen of each sex (NIBRIV0000812791 for the male, and NIBRIV0000812792 for the female; deposited in the National Institute of Biological Resources, Incheon, Korea) was dissected and each part was mounted on a slide glass with lactophenol for further microscopic observation. All measurements were done using an AxioVision LE64 software (AxioVs40x64v 4.9.1.0; Carl Zeiss, Oberkochen, Germany).

For scanning electron microscopy (SEM), adult specimens were dehydrated with absolute ethanol for 15 min. The usual procedure of using a graded ethanol series was skipped since the specimens had been preserved initially in 99.5% ethanol. Sample drying was done using hexamethyldisilazane, HMDS, (CH3)3SiNHSi(CH3)3 (Samchun, Pyeongtaek, Korea) (Braet, De Zanger & Wisse, 1997; Shively & Miller, 2009). Each specimen dehydrated in ethanol was immersed in 1–2 ml HMDS in a 24-well plate, and the plate was placed in a fume hood until the HMDS had totally evaporated. Dried samples were mounted on aluminum SEM stubs and observations were carried out with an S-3000N scanning electron microscope (Hitachi, Tokyo, Japan) operating at an accelerating voltage of 20.0 kV.

Description of morphological characters

Total body length was measured from the anteriormost part of the cephalothorax to the posterior margin of the anal somite. The length of the caudal rami was measured from the inner proximal articulation of a ramus to its most distal point, and width of the rami was measured perpendicular to the length at the level of the insertion of the outermost caudal seta. The terminology for body segmentation used by Grygier & Ohtsuka (2008) was adopted herein.

To describe the antennular setation patterns, the terms and definitions proposed by Grygier & Ohtsuka (1995) and Huys et al. (2007) were mainly used. Distally, however, we found it necessary to propose and define some new descriptive terms in order to help standardize the different terminologies of the two earlier systems. Our schematic diagram of monstrilloid antennular setation is mainly based on eight species of Korean monstrilloids (Fig. 1, Table S1). The highly branched setae “b1–3” and the slightly branched seta “b5” of Grygier & Ohtsuka (1995), which correspond to four of the three-dimensionally branched setae “A–E” of Huys et al. (2007), are here relabeled with upper-case letters, the most dorsal seta as “A”, the two adjacent outer lateral setae as “B” and “C”, and the most distal seta as “D”. These elements are unbranched in some monstrilloid species, but are still distinguishable by their greater length and thickness from the simple setae “b4” and “b6” of Grygier & Ohtsuka (1995), which correspond to the unmodified setal elements “3” and “4” of Huys et al. (2007). These simple setae are labeled herein with lower-case letters, as proximal “a” and distal “b”. The apical spines “61, 2” and inner lateral spine “5” of Grygier & Ohtsuka (1995), which correspond to the unmodified apical spiniform elements “1” and “2” and one inner lateral element “5” of Huys et al. (2007), are labeled herein as “51–3”. The apical aesthetasc, called “6aes” by Grygier & Ohtsuka (1995), is here relabeled as “5aes”. Three long, biserially plumose, strap-like setae called “Vd, Vm and Vv” by Grygier & Ohtsuka (1995) on account of their dorsal, medial, and ventral positions, respectively, are all present in females, but in most males the former two elements are absent and only “Vv” remains, corresponding to setal element “6” of Huys et al. (2007). The more proximal inner minute spiniform element “7” of Huys et al. (2007) is relabeled as “5a” herein. The names of the setal elements on the first to fourth antennular segments generally follow Grygier & Ohtsuka (1995) with the addition of “4a” for a minute spiniform element that has not been mentioned in previous studies. The letter “a” used in the names of the elements “4a” and “5a” means “accessory”, since their consistency of appearance in both males and females and/or in other earlier studied monstrilloids is doubtful. Dorsal element “IVd” on the fourth antennular segment is, in general, present in females, but not in males.

Figure 1 Schematic diagram of basic setal armature of adult monstrilloids.

(A) Updated nomenclatural terms for setal elements of males (left) and females (right), with geniculation indicated by arrow. With respect to the anatomical axes of the antennules, the terms “anterior” and “posterior” pertain to the ancestral, fundamental condition, following Huys et al. (2007). (B) Recommended convention for labeling setal elements on various types of distal antennular segment in male monstrilloids, with apical aesthetasc 5aes serving as reference marker and potential translocation patterns of spiniform elements 51–3 shown by arrows—not to be constructed as an evolutionary hypothesis of character transformation.

Preparation for molecular analysis

Genomic DNA extraction using Chelex® 100 chelating resin (molecular biology grade, 200–400 mesh, sodium form; Bio-Rad, Hercules, CA, USA) was done mainly as in earlier studies (Estoup et al., 1996; Casquet, Thebaud & Gillespie, 2012), but with the final volume reduced to 100–150 μl in order to increase the DNA concentration.

A total of two partial gene sequences, those of mtCOI and 28S rRNA, and the complete gene sequence of ITS1–5.8S–ITS2 were amplified using AccuPower® HotStart PCR PreMix kit (Bioneer, Daejeon, Korea) and thermal cycling was performed using Mastercycler® (Eppendorf, Hamburg, Germany). A total of 20 μl of total reaction volume per reaction tube was prepared by adding 2 μl of DNA template and 1 μl each of forward and reverse primers to 16 μl of distilled water. The primers and the thermal cycling profile for each gene amplification are given in Table 1.

Table 1 Information of primers used for PCR amplifications and thermal cycling profiles.

Gene	Primer	Primer sequence (5′-3′) and thermal cycling profile	Reference	
mtCOI	XcoiF	ATAACRCTGTAGTAACTKCTCAYGC	This study	
HCO2198	TAAACTTCAGGGTGACCAAAAAATCA	Folmer et al. (1994)	
	94 °C, 5 min + [94 °C, 40 s; 50 °C, 45 s; 72 °C, 45 s]40 + 72 °C, 7 min		
ITS1–5.8S–ITS2	ITS5	GGAAGTAAAAGTCGTAACAAGG	White et al. (1990)	
ITS4	TCCTCCGCTTATTGATATGC	White et al. (1990)	
	94 °C, 5 min + [94 °C, 1 min; 48 °C, 1.5 min; 72 °C, 1.5 min]35 + 72 °C, 7 min		
28S rRNA	28S-F1a	GCGGAGGAAAAGAAACTAAC	Ortman (2008)	
28S-R1a	GCATAGTTTCACCATCTTTCGGG	Ortman (2008)	
	94 °C, 5 min + [94 °C, 1 min; 50 °C, 1 min; 72 °C, 1 min]35 + 72 °C, 7 min		
Note:

Primer sequences are given based on the nucleic acid notation formalized by the International Union of Pure and Applied Chemistry (IUPAC).

Amplification of the mtCOI gene by polymerase chain reaction (PCR) was initially attempted using the so-called “universal primers” LCO1490 and HCO2198 (Folmer et al., 1994), but the amplification success rate was generally low. Jeon, Lee & Soh (2018) were able to determine 24 mtCOI gene sequences from 41 monstrilloid individuals, a success rate of just 58.5%. The rate was especially low for Caromiobenella castorea Jeon, Lee & Soh, 2018 (2 of 5 individuals; 40%), Monstrilla grandis Giesbrecht, 1891 (0 of 9; 0%), and Monstrillopsis longilobata (2 of 6; 33%), although the other six studied species had much higher success rates (Jeon, Lee & Soh, 2018: Table S1). For successful PCR in the present study, the internal forward primer XcoiF was newly designed on the basis of the alignment of previously submitted monstrilloid mtCOI gene sequences (Table S2, Fig. S1). Further amplification using the new forward primer and HCO2198 worked properly and resulted in, as expected, a little shorter sequence length in gel electrophoresis, whereas PCR using the universal primers consistently failed (Fig. S2). The present mtCOI sequence products (>500 bp) may be enough to reveal the conspecificity of the present Monstrillopsis species, and also adaptable for further molecular phylogenetic analysis (Blanco-Bercial et al., 2014).

The PCR products were run on a 1% Tris acetate-EDTA agarose gel for 20 min at a voltage of 100 V with a 100 base pair (bp) DNA ladder (Bioneer, Daejeon, Korea). The PCR products with positive results were sent to Macrogen (Seoul, Korea) for purification and DNA sequencing. Sequencing reactions were performed in a DNA Engine Tetrad 2 Peltier Thermal Cycler (Bio-Rad, Hercules, CA, USA) using the ABI BigDye® Terminator v3.1 Cycle Sequencing Kit (Applied Biosystems, Foster City, CA, USA) following the protocols supplied by the manufacturer. Single-pass sequencing was performed on each template using the corresponding primer. The fluorescent-labeled fragments were purified by the method that Applied Biosystems recommends as it removes the unincorporated terminators and dNTPs. For electrophoresis, the samples were injected into an ABI 3730xl DNA Analyzer (Applied Biosystems, Foster City, CA, USA). The sequencing chromatograms were read using FinchTV ver 1.4.0 software. Inspected sequences were taken to the Molecular Evolutionary Genetics Analysis 7 (MEGA7, ver 7.0.21) and then the both forward and reverse primer sites were excluded. The forward and reverse strands were aligned by ClustalW embedded in MEGA7.

Multigene sequences consisting of complete ITS1–5.8S–ITS2 with partial 18S rRNA at the 5′-end and partial 28S rRNA at the 3′-end were compared as a whole, without specific gene region delimitation, since pin-pointing of each gene position by transcript analysis was not attempted. The 5.8S rRNA gene regions in the multigene sequences were, however, estimated by comparing with other complete 5.8S rRNA sequences of copepods from GenBank: Tigriopus californicus (Baker, 1912) (AY599492), Tigriopus japonicus Mori, 1938 (EU057580), Cletocamptus deitersi (Richard, 1897) (AF315025), Ergasilus parasiluri (Yamaguti, 1936) (AY297732), Oithona similis Claus, 1866 (KF153700), Cyclops kolensis Lilljeborg, 1901 (KF153689), and Diacyclops bicuspidatus (Claus, 1857) (KF153697). Doing so eventually allowed the separation of three gene regions: partial 18S rRNA–complete ITS1, 5.8S rRNA, and complete ITS2–partial 28S rRNA.

Nucleotide sequences from three gene regions were newly obtained from six specimens to demonstrate conspecificity between the females and males. Previously submitted nucleotide sequences of M. longilobata for the mtCOI and 28S rRNA genes were also retrieved from GenBank and included in the analysis. In all, eight sequences of mtCOI, nine of ITS1–5.8S–ITS2, and 11 of 28S rRNA were obtained. All gene sequences used for the molecular analysis are listed on Table 2 with accession numbers.

Table 2 List of specimens used for molecular analysis with GenBank accession numbers.

Specimen	Sex	Specimen voucher	GenBank accession number	
mtCOI	ITS1–5.8S–ITS2	28S rRNA	
Monstrillopsis longilobata	Female	HYU-Mon0033	nd	nd	KY563308	
Monstrillopsis longilobata	Female	HYU-Mon0034	nd	MG645220	KY563309	
Monstrillopsis longilobata	Female	HYU-Mon0035	KY553229	MG645221	KY563310	
Monstrillopsis longilobata	Male	HYU-Mon0036	nd	MG645222	KY563311	
Monstrillopsis longilobata	Male	HYU-Mon0037	KY553230	MG645223	KY563312	
Monstrillopsis longilobata	Male	HYU-Mon0038	nd	MG645224	KY563313	
Monstrillopsis longilobata	Female	HYU-Mon0042	MF447158	MG645225	MF447164	
Monstrillopsis longilobata	Female	HYU-Mon0043	MF447159	MG645226	MF447165	
Monstrillopsis longilobata	Female	HYU-Mon0044	MF447160	MG645227	MF447166	
Monstrillopsis longilobata	Male	HYU-Mon0045	MF447161	MG645228	nd	
Monstrillopsis longilobata	Male	HYU-Mon0046	MF447162	nd	MF447167	
Monstrillopsis longilobata	Male	HYU-Mon0047	MF447163	nd	MF447168	
Notes:

Accession numbers for newly obtained sequences presented in bold.

nd, no data.

Systematics

Order Monstrilloida Sars, 1901

Family Monstrillidae Dana, 1849

Genus Monstrillopsis Sars, 1921

Monstrillopsis longilobata Lee, Kim & Chang, 2016

(Figs. 2–12)

Figure 2 Monstrillopsis longilobata Lee, Kim & Chang, 2016, microphotographs of females.

(A) Habitus showing widely spread legs 2–4, dorsal. (B) Habitus showing ventrally projecting ovigerous spines without eggs, lateral. (C) Habitus showing egg mass attached to ovigerous spines, lateral. (D) Habitus with subthoracic brooding, dorsal. Scale bars in micrometer.

Figure 3 Monstrillopsis longilobata Lee, Kim & Chang, 2016, female.

(A) Habitus, dorsal, with right pit-setae 1–14 of right side indicated (cf. Figs. 7A–7D). (B) Habitus, lateral. Scale bars in micrometer.

Figure 4 Monstrillopsis longilobata Lee, Kim & Chang, 2016, female.

(A) Cephalothorax, ventral. (B) Antennule, right, dorsal, setal elements labeled as in Fig. 1A. (C) Detailed of tip of antennule (putative segment 5), right, dorsal, setal elements labeled as in Fig. 1A. (D) Urosome, lateral, showing antero- and posteroventral protuberances of genital compound somite (arrows) and ovigerous spines. (E) Urosome, ventral, showing caudal rami with short ventral setae (arrows) (cf. Fig. 7F). (F) Fifth legs, anterior. Scale bars in micrometer.

Figure 5 Monstrillopsis longilobata Lee, Kim & Chang, 2016, female, swimming legs with intercoxal sclerites.

(A) Leg 1, right, anterior. (B) Leg 2, left, anterior. (C) Leg 3 with well-developed basal seta (arrow), left, anterior. (D) Leg 4, right, anterior. Scale bar in micrometer.

Figure 6 Monstrillopsis longilobata Lee, Kim & Chang, 2016, female, SEM.

(A) Anterior dorsum of cephalothorax showing simple pores (i–v) and band of striations (arrow). (B) Anterior ventral surface of cephalothorax showing scars (S), oral papilla (OP), and pores (black and white arrows). (C) Urosome, dorsal, showing wrinkling on genital compound somite and postgenital somite, pit-setae (13, 14), and simple pores (i, ii). (D) Urosome, lateral, with ventrally projecting ovigerous spines (OS) and posteroventral bulge of genital compound somite (arrow). (E) Fifth leg, left. Scale bars in micrometer.

Figure 7 Monstrillopsis longilobata Lee, Kim & Chang, 2016, female, arrangement of pit-setae (2–14) and urosomal pores (i, ii), SEM.

(A) Incorporated first pediger, right side, dorsolateral, showing closely spaced pit-setae 4 and 5. (B) First free pediger, dorsal, left element 7* is somewhat ambiguous while right side clearly displaying three pit-setae. (C) Second free pediger, dorsal is the presumed site of unseen element 11 on right side marked with asterisk (*). (D) Third free pediger, dorsal. (E) First urosomal somite, dorsal, showing odd number (3) simple pores (i, ii). (F) Caudal rami with extremely short ventral setae (arrows), ventral. Scale bars in micrometer.

Figure 8 Monstrillopsis longilobata Lee, Kim & Chang, 2016, male.

(A) Habitus, dorsal, with right pit-setae 1–14 of right side indicated. (B) Habitus, lateral. Scale bar in micrometer.

Figure 9 Monstrillopsis longilobata Lee, Kim & Chang, 2016, male.

(A) Cephalothorax, ventral. (B) Right antennule, dorsal, with setal elements labeled as in Fig. 1A and inner proximal protuberance (arrow). (C) Urosome, ventral (cf. Fig. 11G). (D) Genital apparatus, dorsal, showing at least three sawtooth-like protuberances at each basal part of genital lappet (arrows). (E) Posterior part of urosome, dorsal (cf. Fig. 12C). Scale bars in micrometer.

Figure 10 Monstrillopsis longilobata Lee, Kim & Chang, 2016, male, swimming legs with intercoxal sclerites.

(A) Leg 1, left, anterior. (B) Leg 2, right, anterior. (C) Leg 3 with well-developed basal seta (arrow), left, anterior. (D) Leg 4, right, anterior. Scale bars in micrometer.

Figure 11 Monstrillopsis longilobata Lee, Kim & Chang, 2016, male, SEM.

(A) Cephalothorax, dorsal, showing dorsal band of striations and crumpled lateral areas (arrows). (B) Anterior dorsum of cephalothorax, dorsal, showing general arrangement of simple pores. (C) Detail of anterior pore group in B, showing pores i and ii. (D) Detail of posterior pore group in B, showing pores iii–v, with presumed site of missing right pore iv marked by asterisk (*). (E) Anterior ventral surface of cephalothorax showing scar (S), oral papilla (OP), and pores (black and white arrows). (F) Urosome, lateroventral, showing lateral striation on first urosomal somite (arrow). (G) Urosome with extremely elongated genital lappets, ventral. (H) Urosomal somites, dorsal, showing area of partial fusion between penultimate somite and anal somite (arrow). Scale bars in micrometer.

Figure 12 Monstrillopsis longilobata Lee, Kim & Chang, 2016, male (A–D) and female (E), SEM.

(A) Incorporated first pediger, left side, dorsal, showing arrangement of pit-setae 1–5. (B) First urosomal somite, dorsal, showing odd number (3) simple pores (i, ii). (C) Caudal rami, dorsal, each armed with four well-developed setae (arrow indicating socket of fellen seta). (D) Legs 4 joined by rectangular intercoxal sclerite (arrow), posterior. (E) Female legs 4 joined by wide intercoxal sclerite (arrow), anterior. Scale bars in micrometer.

Sampling locality

Soho-dong (34°44′50.82″N, 127°39′30.14″E), Yeosu-si, Jeollanam-do, Korea. (English equivalents of political divisions in Korea: dong = village; si = city; do = province).

Material examined

Specimens were collected by using a light trap from 18:31 to 22:59 h at the sampling locality on April 21, 2016. The depth there was about 3 m, and the water temperature cooled from 16.9 to 16.1 °C while the trap was deployed. The female and male specimens used for drawings and measurements are deposited in the National Institute of Biological Resources (NIBR), Incheon, Korea with the following accession numbers: female dissected and mounted on eight slides in lactophenol (NIBRIV0000812792); six intact females in 99.5% ethanol vial (NIBRIV00000812794); sub-mature female in 99.5% ethanol vial (NIBRIV0000812795); male dissected and mounted on eight slides in lactophenol (NIBRIV0000812791); and six intact males in 99.5% ethanol vial (NIBRIV0000812793). Three additional specimens for each sex were used for SEM and deposited in the Laboratory of Zooplankton Diversity, Chonnam National University, Korea. A total of six additional specimens (three females and three males) were utilized for molecular analysis.

Diagnosis (Female)

Total body length 1.48–1.75 mm (mean 1.64; N = 6). Length ratio of cephalothorax, metasome, and urosome 47.9 (range 45.9–49.6): 30.5 (28.1–33.5): 21.6 (19.8–24.6) in lateral view. Metasomal somites brown or dark red except for semi-transparent, bulbous cephalothorax (latter often green due to internal egg mass). Urosomal somites more lightly pigmented than metasomal somites. Anterior dorsum of cephalothorax with narrow band of transverse striations. Two ventral pores between antennular bases. Two prominent scars situated posterior to antennular bases, followed by pair of pores. Faint, incompletely closed reticulation present on part of cephalothorax. Oral papilla moderately developed, located about 21.4% (17.6–24.3%) of way along ventral side of cephalothorax. Two lateral eyes and one ventral eye well-developed and pigmented. Lateral eyes generally bean-shaped in dorsal view, 0.14 mm long, 0.08 mm wide. Ventral eye round, 0.14 mm in diameter. Antennules 2-segmented; second segment formed by incomplete fusion of four subsidiary segments. Antennules 17.3% (16.2–18.7%) as long as total body length. Length ratio of first and second antennular segments 26.8 (25.3–28.6): 73.2 (71.4–74.7). Intercoxal sclerites of leg pairs 2–4 wide and low whereas these of leg pair 1 relatively narrow and high, resulting in widely separated legs 2–4, closely located legs 1 (Figs. 2A and 2D). Ratio of distal width to height of intercoxal sclerites increasing to posterior pairs. Legs of pair 5 also widely separated, bilobed with exopodal lobe carrying three terminal setae, endopodal lobe unarmed, reduced. Urosomal somites with conspicuous longitudinal striations. Genital somite almost completely fused with succeeding somite, forming genital compound somite with prominent dorsal suture; anteroventral part bearing pair of ventrally directed ovigerous spines (Figs. 2B and 2C) 23.0% (21.0–25.3%) as long as total body length. Egg masses carried by ovigerous spines laterally compressed, oval in lateral view, reaching posterior face of first swimming legs (Fig. 2C). Caudal rami subtriangular in dorsal view, each bearing four caudal setae: outer lateral seta arising from midlength of outer margin, two terminal setae, and one remarkably short ventral seta. Inner terminal seta longest, lateral and outer terminal setae subequal in length. Ventral seta reaching only slightly beyond caudal ramus bearing it and hard to distinguish under low magnification.

Diagnosis (Male)

Total body length 1.12–1.28 mm (mean 1.21; N = 7). Length ratio of cephalothorax, metasome, and urosome 45.3 (range 44.5–46.8): 35.2 (32.8–37.0): 19.5 (17.1–20.7) in lateral view. Whole body light brown. Cephalothorax generally cylindrical, slightly constricted from midlength to anterior part of incorporated first pediger. Dorsal transverse striations situated halfway back from anterior end of cephalothorax, reaching to dorsolateral half. Two pores between antennular bases. Two prominent scars situated posterior to antennular bases, followed by pair of pores. Ventral surface from scars to oral papilla transversally striated. Ventral half of cephalothorax bearing narrow, faint band of striations. Oral papilla located anteriorly on ventral surface of cephalothorax, 29.3% (28.1–30.9%) of way back from anterior end. A total of two lateral eyes and one ventral eye well-developed and pigmented. Lateral eyes generally bean-shaped in dorsal view, 0.11 mm long. Ventral eye round, 0.11 mm in diameter, slightly larger than lateral eyes. Antennules 5-segmented with segments 2 and 3 partly fused. Minute spiniform elements 4a and 5a present on antennular segments 4 and 5, respectively. Fifth antennular segment modified with inner hyaline bump and elongated apical spine 52. Antennules 40.0% (38.3–42.9%) as long as body length. Length ratio of antennular segments from proximal to distal 13.9 (12.4–14.8): 20.1 (19.6–21.0): 9.0 (8.1–9.8): 22.4 (21.7–23.2): 34.6 (33.6–35.3). Intercoxal sclerites of leg pairs 1–4 narrow and long, rectangular. Leg pair 5 absent. Fourth free pedigerous somite with longitudinal striations on lateral side. Genital somite bearing robust genital shaft 0.06 mm long and two extremely elongated genital lappets these 0.15 mm long, approximately reaching or slightly exceeding posterior margin of anal somite. Caudal rami subtriangular in dorsal view with four well-developed caudal setae: one outer lateral, two dorso-apical, one ventral.

Description of female, NIBRIV0000812792

Total body length 1.60 mm in dorsal view, 1.68 mm in lateral view. Body consisting of eight somites: cephalothorax incorporating first pedigerous somite, free somites 1–4, genital compound somite, penultimate somite, and anal somite. Length ratio of somites as percentage of total body length 45.9:11.6:11.0:10.6:5.2:9.1:2.4:4.3 in dorsal view, 45.9:11.9:11.3:10.3:5.0:9.3:2.4:3.9 in lateral view. Cephalothorax bulbous (Figs. 3A, 3B and 4A), 0.73 mm long in dorsal view, 0.77 mm in lateral view. Cephalothorax significantly broadening to greatest width of 0.05 mm at 45.0% of its length. At narrowest point, 81.1% of way back, width of waist 0.34 mm. Width of incorporated pediger 0.37 mm at 92.3% length of cephalothorax. Length of metasome including first to third free pediger 0.53 mm in dorsal view, 0.56 mm in lateral view. Length of urosome from first urosomal somite to tip of anal somite 0.33 mm in dorsal view, 0.35 mm in lateral view.

Forehead round with two hair-like sensilla on anterior dorsal surface (Fig. 3A). Anterior fourth of cephalothorax with several pores and striations on dorsal surface. At least four anterior pores recognized, aligned in semi-circle; other pores located slightly behind them (Fig. 6A). Band of transverse striations present starting behind of posterior pore group, not extending onto lateral side (Fig. 3B). Moderately developed oral papilla located ventrally at 17.6% length of cephalothorax, protruding 0.03 mm from ventral surface (Figs. 3B, 4A and 6B). Ventral region halfway from antennular bases to oral papilla with pair of prominent scars followed by transverse striations (Fig. 4A). Ventral pores including anterior pair with subcuticular ducts situated between antennular bases and second pair located closer to midline slightly behind scars (Fig. 6B). Another pair of ventrolateral pores also present (Figs. 3B and 4A).

A total of two lateral eyes and one ventral eye within anterior quarter of cephalothorax, all well-developed, pigmented (Fig. 3B). Lateral eyes round in any direction of view, 147 μm in diameter, situated close together in dorsal view. Ventral eye oval in lateral view, round in dorsal view, 155 μm in diameter, thus slightly bigger than lateral eyes.

Antennules (Figs. 4B and 4C) 2-segmented, not geniculate, directed straight forwards, 0.31 mm long, equaling 40.7% of cephalothorax length, 18.7% of total body length. Length ratio of two segments 27.1:72.9; distal segment evidently formed by incomplete separation of four segments, hereinafter referred to as putative segments 2–5. First segment armed with spine 1 on inner terminal corner. Putative segment 2 armed with five spines (2d1, 2, 2v1–3) and long, strap-like, biserially plumose dorsal seta (IId); ventral spines of 2v series generally longer than dorsal spines of 2d series, 2v3 longest. Putative segment 3 armed with medial spine 3 and IIId and IIIv setae. Putative segment 4 armed with five spines (4d1, 2, 4v1–3), setae IVd and IVv, and ventral aesthetasc (4aes), with 4aes reaching distal margin of antennule. Distal part of antennule (putative segment 5) armed with 13 setal elements: long, strap-like setae Vm, Vd, and Vv, three unmodified spines 51–3, dichotomously branched setae A–D, simple setae a and b, and apical aesthetasc (5aes). Minute setal elements 4a and 5a not observed.

Incorporated first pedigerous somite and first three free pedigers each bearing pair of swimming legs (Figs. 5A–5D and 12E). Protopod consisting of large coxa and small basis separated by diagonal articulation on posterior face and slight indentation on outer margin of anterior face. Seta present on outer margin of each basis, this seta being thin, smooth, and short in legs 1, 2, and 4, reaching approximately to midlength of first exopodal segment, but biserially plumose and much longer in leg 3, reaching to end of exopod (Fig. 5C). Tri-articulate exopod and endopod situated on distal margin of each basis, with endopod always inserted more anteriorly than exopod and shorter than latter, reaching to midlength of third exopodal segment. Setation patterns of swimming legs almost all alike: endopodal segments 1 and 2 each armed with one inner seta, third endopodal segment bearing one outer, two distal, and two inner setae; all endopodal setae biserially plumose, well-developed, subequal in length. Exopodal segment 1 armed with short, robust spine on outer distal corner and short, thin inner seta reaching to about midlength of third endopodal segment. Second exopodal segment bearing well-developed inner seta, outer margin lacking setal elements. Third exopodal segment bearing short, robust spine on outer distal corner plus two terminal and two inner setae on leg 1, two terminal and three inner setae on legs 2–4. Most setae on exopodal segments 2 and 3 biserially plumose, subequal in length, and as long as in endopods, but outermost seta on third exopodal segment serrate along outer margin, uniserially plumose along inner margin. Anterior faces of all third endopodal and exopodal segments with pore (Figs. 5A–5D). Endopodal segments 1–3 fringed along outer margins, exopodal segment 2 also so fringed, but hard to observe by light microscopy. Leg pairs 1–4 all joined by transversally wide trapezoidal sclerites, distal margin (presumably important to secure enough space for subthoracic egg brooding) of which, respectively, 1.8, 2.5, 3.3, and 4.0 times longer than height. Height of each, respectively, 82, 62, 51, and 44 μm. Leg 5 twice as long as wide, members of pair separated at base and widely diverging (Figs. 4F and 6E). Unsegmented protopod dividing into two rami at distal one-third of length. Outer lobe armed with three setae, two of them apical, one at outer distal corner; two outer setae subequal in length, innermost seta thinner and short, all biserially plumose. Inner lobe smooth, unarmed.

Incorporated first pedigerous somite and first three free pedigerous somites with several pairs of pit-setae (sensu Grygier & Ohtsuka, 1995) mainly on dorsal and lateral surfaces (Figs. 3A, 3B and 7A–7D): five pairs (nos. 1–5) on incorporated pediger; three pairs (nos. 6–8) on first free pediger; four pairs (nos. 9–12) on second free pediger; and two pairs (nos. 13, 14) on third free pediger. Pit-setae thin, long. No pit-setae present more posteriorly, but first urosomal somite displaying three simple pores (i, ii) on anterior dorsal margin (Fig. 7E).

First urosomal somite mainly its posterior half and anterior part of anal somite strongly wrinkled longitudinally (Figs. 3B, 4D, 6C and 6D). Anterodorsal part of genital compound somite unwrinkled, but strong, transverse dorsal suture present at midlength of somite. Longitudinal wrinkling present behind suture and continuing onto anterior part of anal somite.

Genital compound somite with swollen anteroventral and posteroventral margins and pair of ovigerous spines arising from anterior ventral surface (Figs. 4D and 6D). These spines directed ventrally perpendicular to body axis, separate basally with distal third thinner than rest, and subequal in length, equal to 25.3% of total body length and 1.2 times longer than urosome.

Pair of caudal rami diverging from posterior part of anal somite. Each ramus 0.11 mm long, 0.06 mm wide, and bearing four caudal setae (Figs. 3A and 4E): one outer lateral, two terminal, one inner ventral, this last seta being remarkably short (Figs. 4E and 7F).

Variation

Specimen (NIBRIV0000812795), carrying unlaid eggs within cephalothorax, with relatively short, corrugated ovigerous spines, seemingly not fully outstretched, only 11.2% of total body length, but still directed ventrally.

Description of male, NIBRIV0000812791

Total body length 1.17 mm in dorsal view, 1.23 mm in lateral view. Body consisting of nine somites: cephalothorax incorporating first pedigerous somite, free somites 1–4, genital somite, post-genital somite, penultimate somite, and anal somite. Last two body somites partly fused, and dorsal articulation represented only by fine suture. Length ratio of somites as percentage of total body length 45.3:11.8:10.6:10.3:4.5:4.9:3.9:3.4:5.4 in dorsal view; 45.3:13.9:11.7:9.6:3.9:4.3:3.4:2.7:5.2 in lateral view.

Cephalothorax cylindrical (Figs. 8A, 8B and 11A), 0.53 mm long in dorsal view, 0.56 mm in lateral view. Greatest width 0.21 mm at half length. At narrowest point, width of waist 0.17 mm. Width of incorporated first pediger 0.21 mm. Metasome 0.38 mm long in dorsal view, 0.43 mm in lateral view. Urosome 0.26 mm long in dorsal view, 0.24 mm in lateral view. Forehead round with two thin sensilla on anterior dorsal surface. Anterior fifth of cephalothorax with several (at least 10) pores (Figs. 8A and 11B–11D): four anterior and six posterior pores, both sets aligned in semi-circle. Dorsal transverse striations at midlength of cephalothorax (Figs. 8A and 11A), reaching to dorsolateral half, forming band-like structure in low magnification (Fig. 8B). Pair of pores with subcuticular ducts situated between antennular bases (Figs. 9A and 11E). Prominent pair of scars situated behind antennular bases (Figs. 9A and 11E). Another pair of pores located slightly behind scars, closer to midline (Fig. 11E). Small oral papilla situated 28.4% of way along ventral side of cephalothorax (Figs. 8B, 9A and 11E). Ventral striations between scars and oral papilla, reaching to ventrolateral half. More ventral striations found at midlength of cephalothorax, these much narrower than dorsal striations (Figs. 9A and 11E).

A total of two lateral eyes and one ventral eye in anterior one-fifth of cephalothorax, all well-developed, pigmented (Figs. 8B and 9A). Lateral eyes round in lateral view, oval in dorsal view, 110 μm in diameter, situated close together. Ventral eye oval in lateral view, round in ventral, 117 μm in diameter, thus slightly bigger than lateral eyes.

Antennules (Fig. 9B) 5-segmented, pointing straight forward, geniculate between fourth and fifth segments, 0.53 mm long, equaling 94.6% of cephalothorax length and 42.9% of total body length. Length ratio of antennular segments from proximal to distal 14.2:19.6:9.7:21.9:34.6. Second and third segments partly fused in dorsal view, clearly separated in lateral and ventral views. Distal antennular segment with crescent-like hyaline bump on inner side at midlength. First antennular segment armed with spine 1 on inner terminal corner. Second segment armed with five spines of subequal length (2d1, 2, 2v1–3) and long, biserially plumose strap-like seta (IId). Third segment armed with spine 3 and plumose IIId and IIIv setae. Fourth segment armed with six spines (4d1, 2, 4v1–3, 4a), seta IVv, and relatively long ventral aesthetasc (4aes); all spiniform elements except 4a biserrate along outer margin. Fifth segment armed with 12 setal elements: spines 51–3, short distal aesthetasc 5aes, dichotomously branched setae A–D, simple and relatively short setae a and b, ventrally located biplumose strap-like seta Vv, and minute spine 5a. Two spines located apically, 51 short, 52 long and robust; 53 located at midlength of fifth segment and moderately developed. Setal elements Vm and Vd absent.

Incorporated first pedigerous somite and three succeeding free pedigers each bearing pair of swimming legs almost identical to those in females (Figs. 10A–10D and 12D). Intercoxal sclerites rectangular, 1.5 times longer than wide, all of almost same width. Leg 5 absent. These four somites also displaying several pairs of pit-setae mainly on dorsal and lateral surfaces (Figs. 8A, 8B and 12A): five pairs (nos. 1–5) on incorporated pediger, with outermost two pit-setae on each side situated close together; three pairs (nos. 6–8) on first free pediger; four pairs (nos. 9–12) on second free pediger; two pairs (nos. 13, 14) on third free pediger. First urosomal somite bearing three simple pores at anterior dorsal margin (Fig. 12B), and longitudinal striations mainly on lateral side (Figs. 8B and 11F). More posterior somites without striations.

Genital somite bearing well developed genital apparatus consisting of robust, 0.07 mm long basal shaft arising from ventral side of somite and two long genital lappets arising from distal corners of shaft (Figs. 9C, 9D, 11F and 11G). Lappets 0.15 mm long, extending beyond end of anal somite, each lappet partly rugose with at least three small, sawtooth-like protuberances at its posterior base (Fig. 9D).

Pair of caudal rami diverging from distal margin of anal somite, 0.08 mm long, 0.05 mm wide, club-shaped with posterior part slightly bulging (Figs. 9C, 9E, 11G and 12C). Each ramus bearing four well-developed caudal setae: one outer lateral, two terminal, one ventral; all setae subequal in length, biserially plumose.

Remarks

The present female specimens have the diagnostic genus-level characters mentioned by Sars (1921): an anteriorly located oral papilla, fully developed eyes, bilobed fifth legs with the outer lobe armed with three setae, a 4-segmented urosome including a genital compound somite, and four setae on each caudal ramus. Some of these features are also mentioned in connection with Monstrillopsis in the keys provided by Davis (1949), Isaac (1975), and Boxshall & Halsey (2004). The most recent generic diagnosis by Suárez-Morales, Bello-Smith & Palma (2006) applied stricter morphological criterion with respect to the number of caudal setae, and it eventually excluded the species with other than four caudal setae from the genus.

A total of six species of Monstrillopsis known from females have been recognized as valid (also see Discussion): Monstrillopsis dubia (Scott, 1904), Monstrillopsis dubioides Suárez-Morales, 2004 (see Suárez-Morales & Ivanenko, 2004), Monstrillopsis ferrarii Suárez-Morales & Ivanenko, 2004, Monstrillopsis chilensis Suárez-Morales, Bello-Smith & Palma, 2006, Monstrillopsis igniterra Suárez-Morales, Ramírez & Derisio, 2008, and Monstrillopsis planifrons Delaforge, Suárez-Morales, Walkusz, Campbell & Mundy, 2017. The cephalothorax of female M. longilobata is distinctly bulbous at its midlength, different from the rather elongate, cylindrical ones of its congeners (cf. Scott, 1904; Sars, 1921; Suárez-Morales & Ivanenko, 2004; Suárez-Morales, Bello-Smith & Palma, 2006; Suárez-Morales, Ramírez & Derisio, 2008, Delaforge et al., 2017). Among the known females, M. dubia and M. dubioides can be instantly excluded from consideration by their size—3.3 mm for M. dubia and 3.8 mm for M. dubioides (Scott, 1904; Suárez-Morales & Ivanenko, 2004)—which far exceeds the mean size of the current female specimens (1.64 mm).

Additional differences mainly concern the relative proportions of the body segments. M. dubia and M. dubioides have a relatively short genital compound somite, respectively, 4% and 4.7% of the body length (Scott, 1904; Suárez-Morales & Ivanenko, 2004), in contrast to 9.3% in M. longilobata. Conversely, the cephalothorax is relatively much longer in the two former species than in the Korean females, comprising over 60% of the body length in the former, but on average 47.9%, and always less than 50%, in M. longilobata. Two other congeners, M. ferrarii (61.3%) and M. igniterra (61.3%), can be also distinguished from the current females the same way (cf. Suárez-Morales & Ivanenko, 2004; Suárez-Morales, Ramírez & Derisio, 2008). The size of the genital compound somite in both species (respectively, 8% and 8.3% of the body length) is similar to, but still slightly shorter than that of M. longilobata (9.3%).

In terms of body proportions, M. chilensis is the closest species to M. longilobata with its cephalothorax constituting 52.3%, and the genital compound somite 9.6%, of the body length (Suárez-Morales, Bello-Smith & Palma, 2006). The total body length (1.76 mm in the holotype of M. chilensis vs. a mean length of 1.64 mm among six females of M. longilobata) and the relative length of the antennules (17.7% and 17.3% of the body length, respectively) are also similar, but other details distinguish the two species. In M. longilobata the inner seta on the first exopodal segment of legs 1–4 is shorter than the respective endopod, reaching to about midlength of the third endopodal segment, whereas in M. chilensis these setae overreach the endopods and reach to about the end of the bearing exopod. Furthermore, the innermost caudal seta of M. chilensis is the shortest, just one-fifth as long as the longest middle seta (Suárez-Morales, Bello-Smith & Palma, 2006), whereas the ventral caudal seta is the shortest in M. longilobata, arising from a small, swollen base and only slightly exceeding the distal end of the caudal ramus. M. longilobata is also distinguished from M. chilensis by the presence of a ventral swelling on the posterior margin of the genital compound somite.

Another congeneric species, the female of M. planifrons which recently reported from the Canadian subarctic region (Dease Strait, Canada), is also different from the present females. The former species is specialized with its species-specific forehead structure of flat, coarse margin and several transverse cuticular ridges at its base (Delaforge et al., 2017). Other prominent differences are presented through the general shape of the cephalothorax (rather elongated, cylindrical in M. planifrons vs. short, bulbous in the present species), and of the fifth legs with the different developmental levels of the inner lobe for each species. The differences on the proportions and the antennular elements are also significant: the present species has shorter antennules (17.3% to the total body length) than those of M. planifrons (31%), the developmental degrees of the antennular spinous elements 61 and 62 (nomenclatures are given as in the original study, and each element corresponds to 52 and 51 of the present study, respectively) of M. planifrons (see Delaforge et al., 2017).

The most striking morphological feature of M. longilobata is the pair of ventrally protruding ovigerous spines, which distinguish the present females from any other known female Monstrillopsis. The pair of ovigerous spines carries laterally compressed egg masses, which are brooded subthoracically in much the same way as those of Maemonstrilla species (see Grygier & Ohtsuka, 2008), but with the anterior margin of the egg mass reaching less far forward and not impinging on the first leg pair. Similar morphological modification of the intercoxal sclerites of legs 2–4, but not that of the first pair of legs, facilitates subthoracic egg brooding inasmuch as the intercoxal sclerites of legs 2–4 are wider than long, thus widely separating the members of each pair, as well as the fifth pair.

The present male specimens obtained from the same samples as the females agree well with the original description of male M. longilobata collected from the eastern coastal waters of Korea, including the possession of extremely long genital lappets: its specific epithet longilobata was originally derived from two Latin words, longus (elongate) and lobatus (lobe-shaped) (Lee, Kim & Chang, 2016). The main difference between the original type series and the current specimens is in the body size: mean 1.26 mm (sum of lengths of body and caudal rami) for the present specimens vs. 1.74 mm for the holotype. In size, the current males resemble Monstrillopsis sarsi Isaac, 1974 (see Isaac, 1974a) at 1.2 mm, but other morphological details more closely resemble those of M. longilobata than those of the latter species. In the original report of Lee, Kim & Chang (2016), M. longilobata was differentiated from M. sarsi by the presence of conspicuous transverse striation on the cephalothorax, a more posteriorly located oral papilla, and very long genital lappets. In addition, the level of modification of the distal antennular “segment” differed. M. sarsi alone was depicted with the distal part of the antennular segment elongate and rather slender; M. longilobata had no such prominent distal elongation.

Scanning electron microscopy provided additional detailed information concerning the pore and pit-seta patterns of the present specimens. The anterior dorsum of the cephalothorax indeed bears two groups of pores as Lee, Kim & Chang (2016) described, but the anterior pores are aligned in a semi-circle as wide as the array of posterior pores, not only aggregated as those authors stated. A total of five pairs of pit-setae are symmetrically situated both laterally and dorsally on the part of the cephalothorax corresponding to the first pedigerous somite, and the most lateral pit-setae (no. 5) and adjacent dorsolateral pit-setae (no. 4) are close to each other. Only one of these pores was depicted in the original description. In general, the number and the location of pores in the examined males coincided well with those of the present females. Both sexes also showed the same kind of ventral band structure consisting of fine transverse striations behind the oral papilla. These striations are finer than the dorsal ones as Lee, Kim & Chang (2016) also noted. The penultimate and anal somites are separated in males by a fine seam on the dorsal surface without prominent arthrodial membrane between them, while the articulation on the ventral side is well defined (Fig. 11H). This feature is also homologous with the previous males reported by Lee, Kim & Chang (2016).

Lee, Kim & Chang (2016) reported the occurrence of males of M. longilobata in southern coastal waters of Korea at Geumo Island, Yeosu, which is close to the current research site at Soho-dong, Yeosu. Morphological similarities and commonality of distribution tend to confirm the conspecificity of the present males with M. longilobata.

Molecular Analysis

The gene sequences were aligned for a length of 523 bp for mtCOI, 794 bp for ITS1–5.8S–ITS2, and 688 bp for 28S rRNA, and the average GC content for each was 30.0%, 44.0%, and 49.0%, respectively. Mean genetic divergence for each group was calculated under the Kimura two-parameter model (K2P) by generating 3,000 bootstrapping replicates. The mean divergence of mtCOI was 0.18% (0.00–0.58%). The mtCOI sequences were, then, translated into amino acid sequences on the basis of an invertebrate mitochondrial genetic code. The amino acid sequences, comprising 174 amino acids (translation starting from the second base), were all identical, without any unexpected internal stop codons. There was 0.15% (0.00–0.38%) mean genetic divergence among the complete ITS1–5.8S–ITS2 sequences from nine individuals. Mean genetic divergences of each 18S–ITS1, ITS2–28S, and 5.8S region were 0.13%, 0.23%, and 0.00%, respectively. Similarly to 5.8S rRNA sequences, there was no genetic divergence at all (0.00%) among the partial 28S rRNA gene sequences from 11 specimens.

Discussion

Systematics of the genus Monstrillopsis

About 20 species of Monstrillopsis have been described globally (Razouls et al., 2005–2018; Walter & Boxshall, 2018). Both sexes of Monstrilla reticulata Davis, 1949 were once assigned to Monstrillopsis by Isaac (1975), but a later study by Suárez-Morales, Bello-Smith & Palma (2006) returned the females to Monstrilla by virtue of the presence of five caudal setae and a seta on the inner lobe of the fifth leg. The males, currently seen as a different species from the females, remained in Monstrillopsis on account of the four caudal setae, modification on the last antennular segment, and the complete absence of the fifth legs (Suárez-Morales, Bello-Smith & Palma, 2006). Another two species, Monstrillopsis angustipes Isaac, 1975 (n. nud.) and Monstrillopsis ciqroi Suárez-Morales, 1993, which have been known only from females, were also excluded from Monstrillopsis due to some morphological discrepancies with respect to their current generic diagnosis of Suárez-Morales, Bello-Smith & Palma (2006). Conversely, Haemocera filogranarum Malaquin, 1901 (= Monstrillopsis filogranarum sensu Suárez-Morales, Bello-Smith & Palma, 2006) has been reallocated to Monstrillopsis, and Monstrillopsis zernowi Dolgopol’skaya, 1948 is tentatively assigned to this genus even though it exhibits an unusual number of caudal setae: six on each caudal ramus in the female, five in the male. An unusual number of caudal setae has been also reported from two other supposedly congeneric males, Monstrillopsis cahuitae Suárez-Morales & Carrillo, 2013 (see Suárez-Morales, Carrillo & Morales-Ramírez, 2013) with six caudal setae, and Monstrillopsis nanus Suárez-Morales & McKinnon, 2014 with five. If the recent generic diagnosis of Monstrillopsis is strictly applied, only 14 species can be recognized as valid: M. dubia, the male of M. reticulata, M. sarsi, Monstrillopsis fosshageni Suárez-Morales & Dias, 2001 (number of caudal setae unknown), M. dubioides, M. ferrarii, M. chilensis, M. igniterra, Monstrillopsis chathamensis Suárez-Morales & Morales-Ramírez, 2009, Monstrillopsis boonwurrungorum Suárez-Morales & McKinnon, 2014, Monstrillopsis hastata Suárez-Morales & McKinnon, 2014, M. longilobata, Monstrillopsis coreensis and M. planifrons. Among these species, three (M. dubia, M. dubioides, and M. chilensis) are known from both sexes, three (M. ferrarii, M. igniterra, and M. planifrons) only from females, and the other eight species only from males.

Another enigmatic species, Monstrillopsis latipes, was introduced in the unpublished doctoral thesis of Isaac (1974b). No later publication or further use of the name followed, and this specific name remains a nomen nudum. Suárez-Morales, Bello-Smith & Palma (2006) brought attention to the unusual ovigerous spines of M. latipes that are “anteriorly directed” and proposed a possible relation with the at that time still undescribed genus Maemonstrilla, the females of which are characterized by anteriorly pointing ovigerous spines. Grygier & Ohtsuka (2008), however, declined to assign this species to Maemonstrilla because the ovigerous spines were described and illustrated by Isaac (1974b) as being directed ventrally, nearly perpendicular to the body axis, not anteriorly, and on account of the lack of information about the intercoxal sclerites and other diagnostic features of the limbs. The ovigerous spines of the present Korean female specimens are also directed ventrally; in some specimens they are inclined slightly anteriorly, but much less so than in Maemonstrilla.

The bilobed fifth legs support the possible assignment of Monstrillopsis latipes to Monstrillopsis, but the number of caudal setae does not. It has been reported to have three caudal setae (Isaac, 1974b; see also Suárez-Morales, Bello-Smith & Palma, 2006), but the actual number may be more. Among the four caudal setae of female M. longilobata, the inner ventral one is very short and thus often hardly recognizable by low magnification light microscopy. Such a seta may well have been overlooked in M. latipes, but if not, assignment of this species to Monstrillopsis may be unsupportable under the current generic diagnosis. Reexamination of the specimen, currently housed in the Natural History Museum, London (unregistered material, labeled Monstrilloida, Jersey. J. Sinel; as stated in Isaac, 1974b) is a priority for further research.

Sexual dimorphism, and male/female matching based on molecular evidence

Several criteria for matching the sexes of monstrilloid species have been used and/or proposed (Gallien, 1934; Grygier & Ohtsuka, 2008; Suárez-Morales, 2011; Lee, Kim & Chang, 2016): co-occurrence of both sexes in a plankton sample or in serial collections conducted over a limited time span at a particular location; the recovery of one form each of both sexes from a single host species; and the sharing of distinctive morphological characters in both sexes. However, each method still carries a high risk of mispairing.

Co-occurrence is the most frequently used matching method, but it is hard to apply with any confidence to samples from places known for high monstrilloid species richness and abundance, such as coral reefs (Sale, McWilliam & Anderson, 1976; Suárez-Morales, 2001; Grygier & Ohtsuka, 2008). The reliability of host specificity is constrained by the lack of much ecological research on host utilization by these copepods. There is currently no guarantee of one-to-one host-parasite specificity in monstrilloids because we cannot completely exclude the case of host sharing by congeneric species with similar morphological structures.

The male specimens of M. longilobata obtained together with the current females display all the typical features of Monstrillopsis. Some of the major morphological features involved are sexually dimorphic: the general shape of the cephalothorax (bulbous in females vs. rather slender and cylindrical in males), the shape of the intercoxal sclerites (wider than long and trapezoidal in females vs. longer than wide and rectangular in males), the detailed morphology of the caudal setae (inner ventral seta short in females), and the presence of sex-specific characters such as the modified distal antennular segment in males, the fifth legs in females, and the totally different genitalia in both sexes. Grygier & Ohtsuka (2008) proposed several species-specific characters of females of Maemonstrilla, for examples, the ornamented coxal lobes of Maemonstrilla polka Grygier & Ohtsuka, 2008 and Maemonstrilla spinicoxa Grygier & Ohtsuka, 2008 and the dorsal spiniform scales of Maemonstrilla turgida (Scott, 1909), that might serve as morphological markers if they are also present in the unknown males of these species (see Grygier & Ohtsuka, 2008). However, among the larger males in their samples they found none that exhibited such features. Our present observation of M. longilobata demonstrated little evidence of morphological similarity between both sexes, and we conclude that matching the sexes by using, or at least solely relying on morphological characters will likely lead to error. Minor features hold some promise. In M. longilobata, the pore patterns (e.g., the general alignment of the integumental organs on the anterior ventral surface of the cephalothorax, the closely adjacent pit-setae 4 and 5 on the dorsolateral side of the incorporated first pediger, and the odd number (three) anterior dorsal pores on the first urosomal somite) and striations (e.g., those between the antennular bases and the oral papilla, and also the dorsal band, agree in both sexes); Lee, Kim & Chang (2016) even predicted that this would be so for the striations. The practical use of such characters to demonstrate conspecificity is, however, restricted by imperfect observations made by light microscopy and the lack of relevant data from earlier studies.

Under these circumstances, the use of molecular techniques is likely to be one of the most efficient and reliable methods for pairing males and females. The molecular analyses presented here support the conspecificity of both sexes of M. longilobata by revealing little or no genetic divergence between them. Hebert, Ratnasingham & de Waard (2003) concluded that mtCOI usually shows about 10% sequence divergence between congeneric species, but that there is a higher, 15.4%, mean divergence in crustaceans. Lefébure et al. (2006) similarly proposed a 0.16 substitution rate per site in the mtCOI sequence as the molecular threshold for species delimitation. Both of the above-mentioned species-delimitation values are much higher than the current mtCOI divergence (mean 0.18%) between male and female M. longilobata. Baek et al. (2016) showed a mean of mtCOI divergence of 2.42% within individual species of copepods, and in particular 1.93% within a species of monstrilloid.

In eukaryotes, rRNA genes are one of the most conserved classes of genes, but they still differ between species (Eickbush & Eickbush, 2007; Rebouças et al., 2013; Zagoskin et al., 2014; Bradford-Grieve, Blanco-Bercial & Boxshall, 2017). In this respect, the 5.8S rRNA gene regions within the present ITS1–5.8S–ITS2 sequences should be also expected to show no difference among individuals of M. longilobata. No transcript analysis for strictly distinguishing and determining any particular gene region was carried out, but we were nonetheless able to estimate the 5.8S rRNA region by aligning and comparing the present gene sequences with the other complete 5.8S rRNA gene sequences registered in GenBank. The comparisons involved at least 155 bp of the 5.8S rRNA region from the present sequences, and the 5.8S rRNA sequences in this position are all homologous in the present six sequenced male and female specimens of M. longilobata. By setting the 5.8S rRNA region as standard, we were also able to distinguish the 18S–ITS1 region (302 bp) and ITS2–28S region (337 bp) because the 5.8S rRNA gene is located between those two regions. The estimated divergences of the 18S–ITS1 and ITS2–28S regions were 0.13% and 0.23%, respectively. These values are probably underestimated because each region contains a small portion of conserved rRNAs (i.e., partial 18S and 28S rRNAs). Precise positioning of the genes would allow more accurate divergence information to be gathered, but our rough estimates at least show that the 5.8S rRNA is highly conserved between the sexes, a fact that supports the conspecificity of the current males and females.

Machida & Tsuda (2010) reported a mean genetic p-distance of less than 0.002 based on “ITS region” within Neocalanus species, but higher values between species: 0.004 for Neocalanus cristatus (Kröyer, 1845) vs. Neocalanus flemigeri Miller, 1988, and 0.007 for N. cristatus vs. Neocalanus plumchrus (Marukawa, 1921). As they defined it, their “ITS region” was equivalent to the present study’s ITS1–5.8S–ITS2–28S complex, so the values derived from their “ITS region” can only be compared with the mean genetic divergence of 0.15% (i.e., 0.0015) derived from the present complete sequence data. This is close to Machida & Tsuda’s (2010) within-species value, and distinctly lower than their between-species values. Krajíček et al. (2016) calculated the mean intra- and inter-genetic divergences of ITS1 sequences from 13 European species of Cyclops. Similarly to the previous study, the mean within-species genetic divergence was 0.26% (calculated based on their Table S8), and the between-species divergence ranged from 3.7% to 20.7%. The purported ITS1 region (i.e., 18S–ITS1) of the present Monstrillopsis species showed a mean genetic divergence of 0.13%, which is much lower than the above-cited between-species values.

Individual consideration of the three gene regions that comprise ITS1–5.8S–ITS2 shows that the genetic differences mainly occurred in the “relatively variable” ITS1 and ITS2 regions but not in the “conserved” 5.8S rRNA region. Similarly, sequences of another form of rRNA, partial 28S rRNA, from 11 individuals of M. longilobata were all identical. In Jeon, Lee & Soh’s (2018) study of 28S rRNA in 11 species of monstrilloids, the sequences within each species group were essentially identical, but they differed between the each species group (21.73% mean divergence). In general, the divergences we found for M. longilobata from these three different gene regions were consistently lower than other reported species-delimitation thresholds or divergence ranges for copepods. This further confirms the conspecificity of the present males and females.

Comparison between females of Monstrillopsis longilobata and Maemonstrilla species

The present females of Monstrillopsis longilobata exhibit an unexpected mixture of morphological characters of the two genera Monstrillopsis as defined by Sars (1921) and Suárez-Morales, Bello-Smith & Palma (2006) and Maemonstrilla as defined by Grygier & Ohtsuka (2008). Monstrillopsis-like features include a moderately developed oral papilla, an inner seta on the first exopodal and endopodal segments of legs 1–4, four caudal setae, and the general shape of the bilobed fifth legs, with the outer lobe bearing three setae and the inner lobe reduced and unarmed; Maemonstrilla-like features include the rather bulbous cephalothorax and the low and wide intercoxal sclerites of legs 2–4 (present on legs 1–4 in females of Maemonstrilla) that appears to be related to subthoracic egg brooding as discussed above. Some characters of Monstrillopsis longilobata are literally intermediate, notably the ventrally directed ovigerous spines (not posteriorly directed as in all other female Monstrillopsis, nor anteriorly directed as in female Maemonstrilla) and the presence on the cephalothorax of relatively prominent transverse striations (as in Monstrillopsis) combined with faint, incompletely closed reticulations that are reminiscent of, but much weaker than the general pattern in Maemonstrilla.

A phylogenetic systematic evaluation of the relationship between Monstrillopsis and Maemonstrilla based on morphological features is rendered more complex by the present females. Maemonstrilla is one of the most clearly defined monstrilloid genera or species-groups and is characterized by unique and complex set of characters. Most of the members (mainly those of the Maemonstrilla hyottoko species group) can be distinguished from the current female Monstrillopsis longilobata by their fifth legs which are long, slender, and rod-shaped with two apical setae, and by the absence of inner setae on the first exopodal and endopodal segments. The Maemonstrilla turgida species group (i.e., M. turgida and Maemonstrilla crenulata Suárez-Morales & McKinnon, 2014) is not quite so distinct. Like the present females, these two species have bilobed fifth legs and an inner seta on the first exopodal and endopodal segments. There are, however, still some differences from M. longilobata: the inner lobe of the fifth leg is armed with a single seta, and the inner setae on the first endopodal segment of legs 1–4 are weakly developed. Furthermore, no dorsal spiniform scales in Maemonstrilla turgida (cf. Grygier & Ohtsuka, 2008: fig. 26B, C) and Maemonstrilla crenulata (cf. Suárez-Morales & McKinnon, 2014: fig. 21A) were detected using SEM in Monstrillopsis longilobata.

The number of caudal setae of Monstrillopsis longilobata is also different from that of Maemonstrilla. In the different monstrilloid genera, variation in the number of caudal setae is quite frequent: five or six in different species of Monstrilla, Maemonstrilla, and Caromiobenella; three or four in different species or sexes of Cymbasoma; and five in Australomonstrillopsis (i.e., Australomonstrillopsis crassicaudata Suárez-Morales & McKinnon, 2014; monotypic) (Huys & Boxshall, 1991; Grygier & Ohtsuka, 2008; Suárez-Morales, 2011; Suárez-Morales & McKinnon, 2014; Jeon, Lee & Soh, 2018). Monstrillopsis displays the widest known range of caudal seta number among the monstrilloid genera, with four to six caudal setae in different species. The majority have four, and this has been regarded as one of the generic characters of Monstrillopsis (sensu Suárez-Morales, Bello-Smith & Palma, 2006), but some species have more: M. zernowi (six in female, five in male), M. cahuitae (six in male; female unknown), and M. nanus (five in male; female unknown). The current study also raises the question of the true phylogenetic relationship between Monstrillopsis and Maemonstrilla, especially the Maemonstrilla turgida species group, as well as the validity of the latter genus. The set of extraordinary features of the female of M. longilobata presents overlaps with both generic diagnoses. It is, however, impossible to come to any conclusion because: (1) no phylogenetic analysis has yet demonstrated whether any morphological characters or character combinations support the monophyly of Monstrillopsis or Maemonstrilla; (2) the unknown males of Maemonstrilla may provide crucial evidence for one classification or another; (3) there are insufficient molecular data to address phylogenetic considerations within the Monstrilloida; (4) the current female Monstrillopsis longilobata itself is disqualified to represent Monstrillopsis in this connection on account of its unusual morphological character set; and (5) a nomenclatural problem is presented by the possible synonymy of Monstrillopsis and Haemocera, the latter having priority (see Jeon, Lee & Soh, 2018).

Considerations on subthoracic egg brooding in the current females

The present females form and carry the egg masses in a manner typical for monstrilloids. They attach the eggs along the ovigerous spines using a mucous substance without forming egg sacs (Malaquin, 1901; Huys & Boxshall, 1991). The general shape of the egg mass is especially similar to egg masses borne by females of Maemonstrilla (Grygier & Ohtsuka, 2008). Subthoracic egg brooding is responsible for this, but one of the most prominent differences between Monstrillopsis longilobata and Maemonstrilla species concerns the anterior extent of the egg mass. In Maemonstrilla, the anterior parts of the large egg masses reach as far forward as the level of the ventral oral papilla on the cephalothorax (see Grygier & Ohtsuka, 2008: fig. 13; Suárez-Morales & McKinnon, 2014: fig. 17). Grygier & Ohtsuka (2008) explained other morphological modifications and adaptions, including the modified intercoxal sclerites of legs 1–4 in Maemonstrilla, for achieving the complete form of subthoracic egg brooding. This explanation is generally applicable to the current females as well, except that the intercoxal sclerite of the first leg pair is relatively narrower and higher than those of leg pairs 2–4. The egg mass, therefore, reaches to just behind the first leg pair, but cannot pass it over forwardly. The ventrally directed ovigerous spines are probably a subsequent adaptation for preventing the physical interruptions between the first leg pair and the egg mass during swimming.

There is also a subtle difference from usual in the shape of the egg mass; in both Monstrillopsis longilobata and Maemonstrilla okame (cf. Grygier & Ohtsuka, 2008: 502) it is laterally compressed with flattened lateral sides. The current females are distinguished from the other species of Maemonstrilla, except for those of the Maemonstrilla turgida species group, by the presence of an inner setae on the first exopodal and endopodal segments. It has been suggested that the absence of the inner setae in other Maemonstrilla species provides room for bearing eggs beneath the thoracic segments (Grygier & Ohtsuka, 2008: 502), but in Maemonstrilla turgida species group and in Monstrillopsis longilobata, their presence might be useful in arranging the eggs and forming the egg mass into the specific shape noted above. Although actual observation in life remains to be done, the inner setae on the first endopodal segments of legs 1–4 may perhaps be involved in evenly distributing the eggs along the ovigerous spines, so as to prevent large parts of the egg mass from falling off and/or minimizing risk from the loss of large egg part at once.

Conclusion

The present study describing females of M. longilobata has led to the following conclusions: (1) the present females of M. longilobata are distinct from any other known congeners of Monstrillopsis in many morphological aspects, especially with the unusually directed ovigerous spines; latter characteristic is also unfamiliar to the genus Monstrillopsis. (2) The presence of severe sexual dimorphism between the females and males of M. longilobata implies that many other monstrilloid species reported from a single sex could possibly have such morphological differences, thus solely relying on the morphological features could be erroneous for a perfect sexual matching. (3) The minor features such as pore patterns and integumental ornamentation, which of those that have not been previously inspected enough for species identification, are more informative than we ever thought, and thus more attention to these features are needed. (4) To prevent further confusion caused by the current usage of two different antennular setal nomenclatures by each sex, those two sets of terms were revised and unified with newly defined terms; the present proposal for the antennular nomenclature well explains the general setal patterns of eight species of Korean monstrilloid copepods. (5) The application of molecular tools is one of the promising methods for compensating the defects caused by insufficient morphological characteristics; newly designed forward primer XcoiF resulted in a little shorter mtCOI sequence product than the case using the “universal primers”, but those sequences are still long enough to confirm a conspecificity of Mn. longilobata. (6) The present females of Mn. longilobata exhibit mixed characteristics of Monstrillopsis and Maemonstrilla, whereas the males are in typical fashion of Monstrillopsis. With the current limited information available, it is insufficient to evaluate the true relationship of two genera, thus more species descriptions and following molecular analyses still remain to be done.

Supplemental Information

Supplemental Information 1 Fig S1. Position of new primer XcoiF and expected sequence length using “XcoiF + HCO2198” primer set.

A total of 24 mtCOI genes covering five genera, and eight species of monstrilloids sharing a conserved region at the base position (bp) of 114th to 138th within the sequences determined using LCO1490 and HCO2198.

Click here for additional data file.

Supplemental Information 2 Fig S2. Agarose gel image showing results of mtCOI gene amplifications.

“XcoiF + HCO2198” primer set results in about 520 bp products as expected (on the left), whereas “LCO1490 + HCO2198” primer set failed proper amplifications (on the right). Four of each female and male specimen used. Specimen numbers given in combination of sex indication (F: female; M: male) and individual distinguishing number; mk: marker (100 bp Ladder, Bioneer).

Click here for additional data file.

Supplemental Information 3 Table S1. Comparison of antennular setal elements of Korean monstrilloids.

Click here for additional data file.

Supplemental Information 4 Table S2. Sequence of new primer and alignment of conserved region of mtCOI gene sequences from eight species of monstrilloids.

Click here for additional data file.

We are grateful to Dr. Mark J. Grygier (Center of Excellence for the Oceans, National Taiwan Ocean University, Taiwan) for providing valuable, inspiring comments, and kind English rewording for overall quality improvements of the manuscript. We also appreciate Prof. Mark J. Costello (Institute of Marine Science, The University of Auckland, New Zealand), Dr. Cristina de Oliveira Dias (Instituto de Biologia, Universidade Federal do Rio de Janeiro, Brazil) and two anonymous reviewers for their constructive comments and advice.

Additional Information and Declarations

Competing Interests

Author Contributions

DNA Deposition

Data Availability

The authors declare that they have no competing interests.

Donggu Jeon conceived and designed the experiments, performed the experiments, analyzed the data, contributed reagents/materials/analysis tools, prepared figures and/or tables, authored or reviewed drafts of the paper, approved the final draft.

Donghyun Lim conceived and designed the experiments, authored or reviewed drafts of the paper, approved the final draft.

Wonchoel Lee conceived and designed the experiments, analyzed the data, authored or reviewed drafts of the paper, approved the final draft.

Ho Young Soh conceived and designed the experiments, analyzed the data, contributed reagents/materials/analysis tools, authored or reviewed drafts of the paper, approved the final draft.

The following information was supplied regarding the deposition of DNA sequences:

(1) The partial sequences of mitochondrial cytochrome c oxidase subunit I described here are accessible via GenBank accession numbers MF447158 to MF447163.

(2) The complete sequences of ITS1–5.8S–ITS2 described here are accessible via GenBank accession numbers MG645220 to MG645228.

(3) The partial sequences of 28S rRNA described here are accessible via GenBank accession numbers MF447164 to MF447168.

The following information was supplied regarding data availability:

The raw data are provided in the Supplemental Files.

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
