# Peer review of "First use of molecular evidence to match sexes in the Monstrilloida (Crustacea: Copepoda), and taxonomic implications of the newly recognized and described, partly Maemonstrilla-like females of Monstrillopsis longilobata Lee, Kim & Chang, 2016"

_PeerJ, doi:10.7717/peerj.4938_

## Round 0.1 · original submission · Minor Revisions

Thank you for this good scientific paper. The referees are unanimously in support of its publication. They also suggest aspects that would improve it, including additional references, corrections and clarifications. One is rightly critical of the (too common) acceptance of genetic data as if it were a gold standard when actually we know as little (perhaps less) about what genetics tell us about what is a valid species as with morhpological characters. I believe there are well distigusihed species with idential CO1 genes for example, as well as some with no morphological but strong COI differences. Please consider these aspects and the referees suggestions in your revision.

Reviewer 1 ·

Basic reporting

This work responds to a need in the taxonomy of monstrilloid copepods that has been repeatedly advanced for many years (Suárez-Morales, 1994, 2005, 2011, 2017), i.e., linking males and females of species by molecular techniques.
The manuscript is very well written, Illustrations are very good, including high-quality line drawings, good half-tone photos and SEM micrographs (SEM-analyzed specimens are remarkably well processed as there is little or no contraction of the cephalothorax cuticle after processing. The taxonomic description is complete and accurate; the characters found in these females are also remarkable, like the interesting, unique postero-ventral process of the genital double-somite, clearly identified and outlined in this work..

Experimental design

The unresolved issues presented in this ms and the lack of alternative explanations (even a moderate speculation) for these questions is my main criticism. The manuscript is well written, authors acknowledged the advice by Dr. Mark J. Grygier, who polished the English text. Illustrations are very good, including high-quality line drawings, good half-tone photos and SEM micrographs (specimens are remarkably well processed as there is little or no contraction of the cephalothorax cuticle after SEM processing. The description is complete and accurate, it follows the highest current standards in copepod taxonomy.

Validity of the findings

Authors opened the discussion about the uncertain status of Maemonstrilla as a valid taxon and suggest that more molecular data will be needed to define thisMonstrillopsis/Maemonstrilla dichotomy It is true that the finding and analysis of Maemonstrilla males will shed more light on the status of these two genera and their relations. Somehow the position they support in this work (i.e.recognizing that minor characters are not enough to link males and females (lines 797-797) contradicts the criteria used elsewhere (Jeon et al., 2018)where a similar set of microcharacters(i.e.,integumental ornamentation, pore /sensilla patterns) was deemed enough morphologic evidence as to establish a new genus of the Monstrilloida. Authors should decide if, based on their exploration of microcharacters among the monstrilloid copepods, these are enough to separate species, genera or link males and females.

Additional comments

An interesting, straightforward contribution that probably generates more questions than answers. Molecules are good tools, but comparative morphology is probably the main source to reach a solid classification within the group, which is clearly more complex than previously thought.The lack of appendages in these copepods hampers the exploration of homologies and the current limitations of molecular techniques will likely conceal their true phylogenetic links for some time.The evidence presented here is solid and well supported by data from different sources; a nice, professional work.

Reviewer 2 ·

Basic reporting

The manuscript is well written, clear and with sufficient context provided. I did enjoy reading the manuscript.

Very straightforward, I have only minor comments.

Experimental design

The experimental design is correct.

Validity of the findings

Data supports conclusions.

Additional comments

Please review (find and replace) M. longilobata by Mn. longilobata in Conclusion. Authors indicate that Monstrillopsis would be Mn. But then break the rule in Conclusion.

When citing support for COI species assignment, they do not cite Blanco-Bercial et al. (2014). It is OK to choose which citations to use for all authors, but this is the most extensive work to date on COI assignment on Copepods.

For the ITS region(s) including 5.8S, authors might want to check Bradford-Grieve et al. (2017). The regions are split, but 5.8s resulted completely conserved in all cases.

In line 719 (“A phylogenetic systematic evaluation…”) might appreciate the clarification that refers to morphology only.

Line 77: “much nucleotide sequence data has” should read “much nucleotide sequence data have”. Data is plural.

Line 178: correct Matercycler to Mastercycler


Blanco-Bercial L, Cornils A, Copley N, Bucklin A (2014) DNA Barcoding of marine copepods: assessment of analytical approaches to species identification. PLOS Currents Tree of Life Edition 1.

Bradford-Grieve JM, Blanco-Bercial L, Boxshall GA (2017) Revision of Family Megacalanidae (Copepoda: Calanoida). Zootaxa 4229, 183.

·

Basic reporting

The Introduction and the Material and Methods show context. Some updates were suggested in the introduction (e.g. lines 54-59).
In the Systematic, the diagnosis is well detailed and described all the taxonomic characterists. The authors could make better use of the figures. In the female and male diagnosis, several taxonomic characteristic was not represented by figures, which are cited only in the description of the sexes (lines 262-313).
In the Molecular Analysis (lines 550-560), was suggested the use of a figure or table in order to illustrate the results.
The literature was well referenced, and the references were relevant. However, the references should be reviewed. There are references that was not cited, and reference that was not listed.
Figures are relevant and with high quality. Some suggestions were made in the labels.
The supplemental material is significantly and relevant.

Experimental design

Research question well defined, relevant and meaningful. The research fills an identified knowledge gap. Methods described with sufficient detail and informations.

Validity of the findings

One novelty is the proposal for the revised nomenclature for antennular setal armament.
The discussion was well substantiated, and the conclusions were according to the objectives.

Additional comments

The main proposals of the authors is demonstrate the conspecificity of individuals of the two sexes of Monstrillopsis longilobata by using both morphological and molecular evidence, and present proposal for the antennular nomenclature explaining the general setal patterns by each sex. The topic is relevant and the purpose of the authors is remarkable. The authors made a meticulous work in relation to the description of taxonomic characters, and they made comparisons with the other known congeners of Monstrillopsis There are, however, some points that the authors should address in order to produce a better version of this manuscript. My principal comments and suggestions are in the pdf file.

---

## Round 0.2 · accepted · Accept

Thank you for the thorough response to the referees reports. I think the figure numbering is ok for the reasons you suggest, and my designation in the Acknowledgements is correct thank you The title is rather long and a more concise version may communicate the novelty better (e.g. Molecular evidence matches sexes in the Monstrilloida (Copepoda) as shown in taxonomy of partly Maemonstrilla-like females of Monstrillopsis longilobata). But I leave final version up the authors.

#